# Fairness under Noise Perturbation: from the Perspective of Distribution Shift

## Abstract

Much work on fairness assumes access to clean data during training. In practice, however, due to privacy or legal concern, the collected data can be inaccurate or intentionally perturbed by agents. Under such scenarios, fairness measures on noisy data become a biased estimation of ground-truth discrimination, leading to unfairness for a seemingly fair model during deployment. Current work on noise-tolerant fairness assumes a group-wise universal flip, which can become trivial during training, and requires extra tools for noise rate estimation. In light of existing limitations, in this work, we consider such problem from a novel perspective of distribution shift, where we consider a normalizing flow framework for noise-tolerant fairness without requiring noise rate estimation, which is applicable to both *sensitive attribute noise* and *label noise*. We formulate the noise perturbation as both group- and label-dependent, and we discuss theoretically the connections between fairness measures under noisy and clean data. We prove theoretically the transferability of fairness from noisy to clean data under both types of noise. Experimental results on three datasets show that our method outperforms state-of-the-art alternatives, with better or comparable improvements in group fairness and with relatively small decrease in accuracy under single exposure and the simultaneous presence of two types of noise.

## 1 Introduction

As machine learning systems are increasingly used in high-stake social areas, there have been arising concerns that automatic decision-making systems, if not properly regulated or intervened, would perpetuate or amplify existing biases and discrimination in society (Angwin et al., 2016; Dressel and Farid, 2018; De-Arteaga et al., 2022; Ricci Lara et al., 2022). It has been shown that merely removing sensitive information during training is not sufficient to ensure fairness, as there may be correlation or causality between sensitive attributes and other features used in the training process, which could result in discriminatory outcomes (Jackson, 2018; Mehrabi et al., 2021). In response, different metrics and methods on fairness (Hardt et al., 2016; Zafar et al., 2017; Choi et al., 2020; Diana et al., 2022) have been proposed to quantify discrimination and to achieve parity for machine learning models.

Current literature on fairness generally assumes access to full and clean sensitive information when imposing fairness intervention. In practice, however, due to privacy or legal concern, it is sometimes infeasible to collect or use such information, greatly hindering the application of conventional methods on fairness (Lahoti et al., 2020; Chai et al., 2022); moreover, the collected sensitive information can be subject to noisy perturbation, leading to inaccurate estimation of unfairness (Fioretto et al., 2022). Despite recent works on proxy sensitive attribute (Yan et al., 2020; Grari et al., 2021), it has been shown that noisy protected information alone, without extra regulation, is not a sufficient

substitution for ground-truth sensitive information (Lamy et al., 2019). Therefore, it is crucial to study the problem of fairness under noisy sensitive information.

Much of current work on fairness under noisy sensitive information requires access to noise rate or external tools for noise rate estimation and uses group-dependent noise rate to rectify measures of unfairness during training (Wang et al., 2020; Celis et al., 2021; Mehrotra and Celis, 2021). However, the estimation process can be costly and inaccurate up to varied estimation methods, and such formulations may not work well under varying noise rates between training and testing data. Besides, much of current formulation regarding noisy sensitive information assumes uniform flip within different groups, which in return, could lead to trivial modifications of fairness constraints during training, especially in terms of complex neural networks. Instead, we seek to find alternative ways to quantify disparities and to improve fairness under noisy sensitive information, without using extra tools for noise evaluation.

We draw inspirations from fairness under distribution shift, where the goal is to ensure the transferability of fairness and accuracy between source (training) distribution and target (testing) distribution (Rezaei et al., 2021; Singh et al., 2021) for a given classifier. Specifically, in terms of noisy sensitive information, we can readily think of the noisy distribution as source, and clean distribution as target. However, most work on distribution shift requires access to the target distribution, which in return, requires external tools for noise rate evaluation.

In light of current limitations in both aspects, in this work, we propose a general framework for fairness under noisy sensitive attribute from the perspective of distribution shift. We consider group- and class-dependent noise rates within each subgroup, and we show that under such formulation, fairness metrics under noisy attributes are not necessarily proportional to those under clean attributes. We propose to solve the problem from the perspective of fair representation learning, where the idea is to train a fair encoder such that its latent representation achieves desired fairness and accuracy properties. We quantify disparities between noisy and clean distributions from the perspective of group- and class-dependent distribution shift under our formulation of noisy sensitive information, and we show theoretically that under bounded divergence between noisy distributions of different subgroups, we have the transferability of fairness guarantee between noisy and clean data, where disparities under clean data are upper-bounded by disparities under noisy data up to addictive and multiplicative constants. In this way, we are able to achieve fairness under noisy protected information, without applying extra techniques for noise rate estimation. What's more. we extend our method to fairness under label noise, where we show both theoretically and experimentally that our method improves fairness under group- and label-dependent label noise.

We summarize our contribution as follows:

1. We discuss two types of noise (i.e., sensitive attribute noise and label noise) under group- and label-dependent assumptions, and we derive the theoretical connections between fairness measures under noisy and clean data in the presence of each type of noise.

2. We formulate fairness under sensitive attribute noise through a novel perspective of distribution shift, from which we introduce a representation learning framework without requiring extra techniques for noise rate estimation. Moreover, we extend our framework to address fairness under label noise.

3. We prove theoretically the transferability of fairness between noisy and clean data both under sensitive attribute noise and label noise.

4. We validate the effectiveness of our method in improving fairness through experiments on three benchmark datasets, where we evaluate its performance under both single exposure and simultaneous presence of sensitive attribute and label noise.

## 2   Related work

**Fairness in Machine Learning**: Discrepancies in machine learning systems against certain groups or subgroups are generally considered to be originated from biased training data, rather than the training process (Kleinberg et al., 2016). To quantify such disparities, different fairness notions have been proposed, including disparate impact (Willborn, 1984), equal opportunity and equalized odds Hardt et al. (2016), Lipschitz continuity (Dwork et al., 2012; Yurochkin et al., 2019) and calibration(Dwork et al., 2012) for individual fairness. Accordingly, different methods have been

proposed to mitigate bias during the training process. Preprocessing methods (Tan et al., 2020; Li and Liu, 2022; Kleindessner et al., 2023) aims at obtaining a rectified distribution of input features or labels such that the desired fairness measures are satisfied on the training set. Inprocessing methods (Madras et al., 2018; Roh et al., 2020; Chai et al., 2022) aim at reagulating the training process with relaxed fairness constraints. Postprocessing methods aim at adjusting decision thresholds (Hardt et al., 2016; Corbett-Davies et al., 2017; Hsu et al., 2022) for each group or learning a instance-wise mapping of soft labels based on expected fairness measures. However, most of existing work on fairness is formulted without considering the effect of label or attribute noise.

**Noise-Tolerant Fairness**: Existing work on fairness under attribute noise relies on the estimation of noise rates. Lamy et al. (2019) first proposes a general framework for fairness under group-dependent attribute noise, and propose to rectify unfairness tolerance during training based on noise rate estimation. Celis et al. (2021) considers the problem similarly by rectifying fairness constraints during training with niose transition matrix. Wang et al. (2020) considers the problem from the perspective of distributionally robust optimization and uses soft group assignment to rectify fairness constraint. Mehrotra and Celis (2021) proposes a preprocessing framework based on sample selection with relaxed weight constraints specified by noise rates.

Methods on fairness under label noise generally focuses on rectifying fairness measures based on estimated noise rates. Work including (Wang et al., 2021; Wu et al., 2022) proposes to replace fairness constraints on noisy data with their corresponding surrogate measures on clean data. Zhang et al. (2023) proposes a VAE-based framework to achieve disentanglement between input feature and sensitive information and uses mutual information between noisy and clean label as penalty term.

**Fairness under Distribution Shift**: Distribution shift has been shown to be non-trivial in fairness and could significantly deteriorate discrimination of a fair classifier (Mishler and Dalmasso, 2022; Schrouff et al., 2022; Chai and Wang, 2023). A general assumption in distribution shift is that labelled source distribution $(X, Y, A) \sim P_{src}$ and unlabelled target distribution $(X, A) \sim P_{trg}$ are accessible during training. Generally, methods on fairness under distribution shift falls into two categories: importance reweighting (Sugiyama et al., 2007; Cortes et al., 2010), where the idea is to reweight instance-wise training loss based on the corresponding ratio between source and target distribution, and robust log loss (Rezaei et al., 2020; Singh et al., 2021; Rezaei et al., 2021; An et al., 2022), where the idea is to formulate training problem as a mini-max optimization problem with robust training loss. Chen et al. (2022) proposes to quantify transferability of fairness under bounded distribution shift represented by group-wise shift vectors, where feature shift and label shift are considered separately.

## 3 Method

Throughout this section, we use $mea$ to denote measures under clean data, $\hat{mea}$ and $\widetilde{mea}$ to denote measures under sensitive attribute noise and under label noise, respectively. For example, we use $\{A, Y\}$ to denote the random variables of sensitive attribute and label under clean data, $\{\hat{A}, Y\}$ the random variables under attribute noise, and $\{A, \widetilde{Y}\}$ the random variables under label noise. We use $\eta$ and $\beta$ to denote sensitive attribute noise rate and label noise rate, respectively.

### 3.1 Problem Formulation

Let $\{(x_i, y_i, a_i), 1 \leqslant i \leqslant N\}$ be the training set where $x_i \in \mathbb{R}^n$ is the input feature, $y_i \in \{0, 1\}$ the training label and $a_i \in \{0, 1\}$ the sensitive attribute, let $f$ be the function of classifier, a general fair classification problem can be formulated as

$$\arg\min_f \frac{1}{N} \sum_{i=1}^{N} l(f(x_i), y_i), \text{ s.t. } l_f(f(x_i), y_i, a_i) \leqslant \epsilon,$$

where $l$ is the classification loss and $l_f$ is the fairness constraint specified by designated fairness notions. For example, $l_f = |\frac{\sum_{\{i|a_i=a\}} \mathbb{1}[f(x_i) \geq 0.5]}{|\{i|a_i=a\}|} - \frac{\sum_{\{i|a_i=a'\}} \mathbb{1}[f(x_i) \geq 0.5]}{|\{i|a_i=a'\}|}|$, where $a' = |1-a|$, corresponds to disparate impact (DI), and $l_f = |\frac{\sum_{\{i|a_i=a,y_i=0\}} \mathbb{1}[f(x_i) \geq 0.5]}{|\{i|a_i=a,y_i=0\}|} - \frac{\sum_{\{i|a_i=a',y_i=0\}} \mathbb{1}[f(x_i) \geq 0.5]}{|\{i|a_i=a',y_i=0\}|}| +$ $|\frac{\sum_{\{i|a_i=a,y_i=1\}} \mathbb{1}[f(x_i) \geq 0.5]}{|\{i|a_i=a,y_i=1\}|} - \frac{\sum_{\{i|a_i=a',y_i=1\}} \mathbb{1}[f(x_i) \geq 0.5]}{|\{i|a_i=a',y_i=1\}|}|$ corresponds to equalized odds (EOd).

In the presence of sensitive attribute noise, such formulation can result in a biased estimation of discrimination on training data. Previous work (Lamy et al., 2019; Celis et al., 2021) has shown that under group-dependent sensitive attribute noise rate $\eta_a := p\left[A \neq \hat{A} | \hat{A} = a\right]$, fairness measure under noisy data is proportional to that under clean data:

$$\hat{l}_f = (1 - \eta_a - \eta_{a'})l_f.$$

However, such formulation can become trivial during training, especially for deep neural networks, where different noise rates can become ignorable under hyperparameter-tuning due to the proportionality. Instead, we consider a more general version of attribute flip, where noise rates are both group-dependent and label-dependent. Specifically, let $P_{ya}$ and $Q_{ya}$ be the distribution of data and predicted soft labels in the clean subgroup $\mathbb{S}_{ya} := \{i | y_i = y, a_i = a\}$ respectively, let $\eta_{ya} := p\left[A \neq \hat{A} | Y = y, A = a\right]$ be the sensitive attribute noise rate in the corresponding subgroup, we have the following relationship regarding noisy and clean distribution:

$$\hat{P}_{ya} = (1 - \eta_{ya})P_{ya} + \eta_{ya'}P_{ya'}. \tag{1}$$

Correspondingly, we have the following relationship regarding DI and EOd under clean and noisy data:

**Lemma 1.** *Under group- and label-dependent attribute noise rate $\eta_{ya}$, we have*

$$\hat{EOd} = (1 - \eta_{10} - \eta_{11})DTPR + (1 - \eta_{00} - \eta_{01})DTNR,$$

$$\hat{DI} = |\lambda_0 FPR_0 - \lambda_1 FPR_1 + (\hat{\alpha}_0 - \hat{\alpha}_0\eta_{10} - \hat{\alpha}_1\eta_{11})TPR_0 - (\hat{\alpha}_1 - \hat{\alpha}_0\eta_{10} - \hat{\alpha}_1\eta_{11})TPR_1|,$$
$$\lambda_a = [1 - (\hat{\alpha}_a + \eta_{0a}) + \hat{\alpha}_a\eta_{0a} - \eta_{0a'} + \hat{\alpha}_{a'}\eta_{0a'}],$$

where DTPR (disparate true positive rate) $=$ $|TPR_0 - TPR_1|$ is the difference in true positive rate (TPR) between the two sensitive groups $\{i | a_i = 0\}$ and $\{i | a_i = 1\}$, DTNR (disparate true negative rate) $=$ $|TNR_0 - TNR_1|$, and $\hat{\alpha}_a = \frac{|\{i | \hat{a}_i = a, y_i = 1\}|}{|\{i | \hat{a}_i = a\}|}$ is the base rate of noisy data at group $\{i | \hat{a}_i = a\}$. Here we assume $\eta_{ya} + \eta_{ya'} \leqslant 1$; for $\eta_{ya} + \eta_{ya'} \geqslant 1$, it is easy to come to equivalent expressions due to symmetry. From Lemma 1, we observe that under group- and label-dependent noise, EOd under noisy data can be expressed as a weighted sum of disparate TPR and TNR under clean data, while DI under noisy data takes a more complicated form involving both noisy base rates and noise rates and does not have a similar relationship with DI under clean data due to possible change in base rates. Correspondingly, optimizing over DI or EOd directly on noisy data may not lead to satisfying improvement in fairness, if without noise estimation.

## 3.2 From the Perspective of Distribution Shift

Estimation of sensitive attribute noise can be inaccurate. Instead, we aim to find a general way for fairness under attribute noise without using extra tools for noise estimation. Note from Lemma 1 that the deviation in fairness measure under noisy data is, in fact, induced by the disparities between noisy and clean distribution, leading to skewed estimation of group-wise utilities. Thus, one direct implication is to consider the problem from the perspective of *covariate shift* on training set. Specifically, we have the clean distribution of data as weighted subtraction of noisy distributions:

$$P_{ya} = \frac{1 - \eta_{ya'}}{1 - \eta_{ya} - \eta_{ya'}}\hat{P}_{ya} - \frac{\eta_{ya}}{1 - \eta_{ya} - \eta_{ya'}}\hat{P}_{ya'}. \tag{2}$$

Consider noisy data as the source distribution and clean data as target, we have the KL-divergence between noisy and clean distribution at each subgroup as follows:

$$D_{KL}(\hat{P}_{ya}||P_{ya}) = \int \hat{P}_{ya} \log \frac{\hat{P}_{ya}}{P_{ya}} = -\int \hat{P}_{ya} \log \left[ \frac{1 - \eta_{ya'}}{1 - \eta_{ya} - \eta_{ya'}} - \frac{\eta_{ya}\frac{\hat{P}_{ya'}}{\hat{P}_{ya}}}{1 - \eta_{ya} - \eta_{ya'}} \right]. \tag{3}$$

This indicates that the discrepancy, or shift between noisy and clean distribution are in fact, controlled by the discrepancy between corresponding noisy subgroups. By minimizing the divergence between

data distribution $\hat{P}_{ya}$ and $\hat{P}_{ya'}$, which, in return, minimizes the divergence between predicted soft label distribution $Q_{ya}$ and $\hat{Q}_{ya}$ and thus provides fairness guarantee for noisy data, we are able to minimize the divergence between noisy and clean distribution. Therefore, by minimizing the divergence between $\hat{P}_{ya}$ and $\hat{P}_{ya'}$ we are able to ensure the transferability of fairness improvement between noisy and clean data. Specifically, when $\hat{P}_{ya} = \hat{P}_{ya'}$, we have $D_{KL}(\hat{P}_{ya}||P_{ya}) = 0$.

## 3.3 Fair Representation Learning

Inspired by Eq. (3), transferability of fairness between clean and noisy data can be ensured under equalized distribution on noisy data: $\hat{P}_{ya} = \hat{P}_{ya'}$, $\forall y$. Due to disparities on training data, however, such requirement is generally infeasible without applying extra regularization. Therefore, we consider a fair representation learning method for fairness under noisy attribute based on normalizing flow. Let $g_{ya}$ be the function of bijective encoder for samples in the noisy subgroup $\hat{\mathbb{S}}_{ya}$ and $h$ be the function of classification head, let $z_{ya} = g_{ya}(x)$ be the latent representation of the corresponding subgroup and $P_{z_{ya}}$ be the corresponding density, we can use change of variables formula to calculate the densities of $z_{ya}$ as:

$$\log P_{z_{ya}}(\boldsymbol{z}) = \log P_{ya}\left(g_{ya}^{-1}(\boldsymbol{z})\right) + \log \left|\det \frac{\partial g_{ya}^{-1}(\boldsymbol{z})}{\partial \boldsymbol{z}}\right|. \tag{4}$$

Following Balunović et al. (2021), we use symmetrized KL-divergence to approximate the statistical distance between subgroups:

$$\mathcal{L}_y = \frac{1}{B}\sum_{j=1}^{B}\left(\log P_{z_{ya}}\left(\boldsymbol{z}_{ya}^j\right) - \log P_{z_{ya'}}\left(\boldsymbol{z}_{ya}^j\right) + \log P_{z_{ya'}}\left(\boldsymbol{z}_{ya'}^j\right) - \log P_{z_{ya}}\left(\boldsymbol{z}_{ya'}^j\right)\right), \forall y \tag{5}$$

where $B$ is the batch size. And the overall training objective can be written as

$$\underset{g_{00},g_{01},g_{10},g_{11},h}{\arg\min} \lambda_0\mathcal{L}_0(g_{00},g_{01}) + \lambda_1\mathcal{L}_1(g_{10},g_{11}) + (1 - \lambda_0 - \lambda_1)\mathcal{L}_{cls}(g_{00},g_{01},g_{10},g_{11},h). \tag{6}$$

## 3.4 Theoretical Analysis

It is easy to see from Eq. (2) that when $\hat{P}_{ya} = \hat{P}_{ya'}$, we also have $P_{ya} = P_{ya'}$ regardless of noise rates, and the classifier achieves perfect EOd on both clean and noisy data. In reality, however, it is hard to achieve prefect fairness. The following theorem states a general relationship between fairness measure under clean and noisy data:

**Theorem 1.** *Let $Q_{ya}$ and $\hat{Q}_{ya}$ be the distribution of predicted soft labels in the clean subgroup $\mathbb{S}_{ya}$ and noisy group respectively, let $\eta_{ya} := p\left[A \neq \hat{A}|Y = y, \hat{A} = a\right]$ be the group- and class-dependent noise rate. For $D_{KL}(\hat{Q}_{ya}, \hat{Q}_{ya'}) \leq \epsilon_y$, we have the following upper- and lower-bound regarding EOd under clean distribution and $\hat{EOd}$ under noisy distribution:*

$$\hat{EOd} \leq EOd \leq \hat{EOd} + \frac{\eta_{00} + \eta_{01}}{1 - \eta_{00} - \eta_{01}}\sqrt{\epsilon_0} + \frac{\eta_{10} + \eta_{11}}{1 - \eta_{10} - \eta_{11}}\sqrt{\epsilon_1}. \tag{7}$$

We defer full proof to appendix. Theorem 1 shows that, despite $\hat{EOd}$ itself serves as a biased estimation of ground-truth EOd, by minimizing the KL-divergence between distributions of soft labels in label-wise subgroups, we are able to minimize the upper-bound of clean EOd, which validates the feasibility of our method.

## 3.5 Fairness under Noisy Labels

We further extent our method to fairness under noisy labels. Following previous work on fairness under label noise (Wang et al., 2021), we consider group- and label-dependent noise rates. Let $\beta_{ya} := p\left[Y \neq \widetilde{Y}|\widetilde{Y} = y, A = a\right]$ be the label noise rate at the subgroup $\widetilde{\mathbb{S}}_{ya}$, we have the following relationship regarding the distribution of clean and noisy data:

$$\widetilde{P}_{ya} = (1 - \beta_{ya})P_{ya} + \beta_{ya}P_{y'a}, \tag{8}$$

where $y' = |1 - y|$. Correspondingly, we have the following relationship regarding fairness measures under clean and noisy data:

**Lemma 2.** *Under group- and label-dependent label noise rate $\beta_{ya}$, we have*

$$\widetilde{DI} = DI,$$

$$\begin{aligned}
\widetilde{EOd} =& |TPR_0 - TPR_1 + \beta_{10}(FPR_0 - TPR_0) - \beta_{11}(FPR_1 - TPR_1)| \\
&+ |TNR_0 - TNR_1 + \beta_{00}(FNR_0 - TNR_0) - \beta_{01}(FNR_1 - TNR_1)|,
\end{aligned}$$

which shows that under label noise, $\widetilde{DI}$ itself serves as an unbiased estimation, while $\widetilde{EOd}$ is not an unbiased estimation of EOd. A natural question here is, *does our method also work under label noise?* The following lemma shows the connection between $\widetilde{EOd}$ and EOd:

**Lemma 3.** *let $\beta_{ya}$ be the group- and class-dependent label noise rate, we have the following upper-bound regarding EOd under clean distribution and $\widetilde{EOd}$ under noisy distribution:*

$$EOd \leqslant \min\left\{\frac{1}{1 - \beta_{00} - \beta_{10}} + \beta, \frac{1}{1 - \beta_{01} - \beta_{11}} + \beta\right\} \widetilde{EOd} + 2\beta,$$

$$\beta = \max\left\{\left|\frac{\beta_{00}}{1 - \beta_{00} - \beta_{10}} - \frac{\beta_{01}}{1 - \beta_{01} - \beta_{11}}\right|, \left|\frac{1 - \beta_{00}}{1 - \beta_{10} - \beta_{00}} - \frac{1 - \beta_{01}}{1 - \beta_{11} - \beta_{01}}\right|\right\}.$$

We defer full proof to appendix. Therefore, under label noise, optimizing over $\widetilde{EOd}$ can still benefit EOd, which is upper-bounded by $\widetilde{EOd}$ up to an multiplicative constant and an addictive constant determined by the noise rates.

# 4 Experiments

## 4.1 Experimental Setup

We validate our method on three benchmark datasets: **COMPAS**: The COMPAS dataset (Larson et al., 2016) contains 7,215 samples with 11 attributes. Following previous works on fairness (Zafar et al., 2017), we only select black and white defendants in COMPAS dataset, and the modified dataset contains 6,150 samples. The goal is to predict whether a defendant reoffends within two years, and we choose *race* as sensitive attributes. **Adult**: The Adult dataset (Dua and Graff, 2017) contains 65,123 samples with 14 attributes. The goal is to predict whether an individual's income exceeds $50K$, and we choose *gender* as sensitive attributes. **CelebA**: The CelebA dataset (Liu et al., 2015) contains 202,599 face images, each of resolution $178 \times 218$, with 40 binary attributes. We choose gender as labels and *age* as sensitive attributes.

We implement our method in PyTorch 2.0.0 on one RTX-3090 GPU. We use accuracy as utility measure, DI (Willborn, 1984) and EOd (Hardt et al., 2016) as fairness measure. We use RealNVP (Dinh et al., 2016) to build our models, and network structures for other methods are chosen as MLP for COMPAS and Adult datasets, and ResNet-18 for CelebA dataset. We repeat experiments on each dataset three times and report the average results. In each repetition, we use a 80%-20% training-testing partition of data.

We compare our method with following related methods:

- **Baseline**: Neural network without fairness regularization.
- **Inprocessing**: Neural network with relaxed EOd constraint by (Wang et al., 2022). This is a fairness method without considering noisy data.
- **DLR**: Neural network with fairness constraints rectified by noise transition matrix (Celis et al., 2021). This method focuses on fairness with noisy sensitive attributes.
- **FairExpec**: Neural network with instance-wise reweighing as specified by (Mehrotra and Celis, 2021). This method focuses on fairness with noisy sensitive attributes.
- **CorScale**: Neural network with rectified fairness constraints (Lamy et al., 2019). This method focuses on fairness with noisy sensitive attributes.
- **SurrogateLoss**: Neural network with modified EOd constraint by (Wang et al., 2021). This method focuses on fairness with noisy labels.

## 4.2 Fairness with Noisy Protected Attributes

### 4.2.1 Fairness under given noise rates

Results on fairness under noisy sensitive attributes are shown in Table 1-3. Compared with baseline and inprocessing which also do not require estimation of noise rates our method achieves a better trade-off in terms of fairness and accuracy, with significant improvement in fairness with smaller or comparable sacrifice in accuracy. Compared with methods that require noise estimation (DLR, FairExpec and CorScale), our method achieves better or comparable performance in terms of both fairness and accuracy, which validates the effectiveness of our method.

| Method | Accuracy | Disparate Impact | EOd |
|---|---|---|---|
| Baseline | 66.80±0.34% | 24.13±1.46% | 42.96±2.02% |
| Inprocessing (Wang et al., 2022) | 62.35±0.65% | 13.34±1.15% | 17.68±1.47% |
| DLR (Celis et al., 2021) | 60.34±0.79% | 11.26±1.35% | 10.46±1.89% |
| FairExpec (Mehrotra and Celis, 2021) | 62.27±1.18% | 10.36±1.27% | 12.26±1.52% |
| CorScale (Lamy et al., 2019) | 61.37±0.68% | 15.25±1.26% | 21.37±2.21% |
| Ours | 63.65±0.87% | 9.94±1.45% | 8.67±2.95% |

Table 1: Experimental results on COMPAS dataset under sensitive attribute noise. The noise rates are set as $\eta_{00} = 0.2$, $\eta_{01} = 0.1$, $\eta_{10} = 0.3$, $\eta_{11} = 0.2$.

| Method | Accuracy | Disparate Impact | EOd |
|---|---|---|---|
| Baseline | 84.16±0.45% | 16.67±1.35% | 20.27±1.13% |
| Inprocessing (Wang et al., 2022) | 82.27±0.69% | 13.34±1.58% | 16.29±1.53% |
| DLR (Celis et al., 2021) | 78.67±0.66% | 9.64±1.35% | 11.17±1.28% |
| FairExpec (Mehrotra and Celis, 2021) | 81.65±0.59% | 9.94±1.45% | 12.28±1.13% |
| CorScale (Lamy et al., 2019) | 80.27±0.45% | 11.96±1.12% | 14.57±1.86% |
| Ours | 82.11±0.64% | 9.97±1.32% | 6.84±1.59% |

Table 2: Experimental results on Adult dataset under sensitive attribute noise. The noise rates are set as $\eta_{00} = 0.15$, $\eta_{01} = 0.1$, $\eta_{10} = 0.1$, $\eta_{11} = 0.3$.

| Method | Accuracy | Disparate Impact | EOd |
|---|---|---|---|
| Baseline | 89.43±0.57% | 22.69±1.86% | 18.32±1.67% |
| Inprocessing (Wang et al., 2022) | 86.47±0.83% | 16.49±1.52% | 15.21±1.46% |
| DLR (Celis et al., 2021) | 86.27±0.62% | 12.54±1.76% | 11.58±1.29% |
| FairExpec (Mehrotra and Celis, 2021) | 85.54±0.69% | 11.45±1.84% | 11.27±1.65% |
| CorScale (Lamy et al., 2019) | 85.34±1.17% | 14.26±1.33% | 13.16±1.58% |
| Ours | 87.14±0.68% | 8.84±1.42% | 8.43±1.19% |

Table 3: Experimental results on CelebA dataset under sensitive attribute noise. The noise rates are set as $\eta_{00} = 0.1$, $\eta_{01} = 0.2$, $\eta_{10} = 0.3$, $\eta_{11} = 0.1$.

### 4.2.2 Fairness under Varying Noise Rates

We move on to discuss results on fairness under varying noise rates. Specifically, we use noise rates in previous sections as baseline rates and vary each component within the range of $[0, 0.5]$. Results under varying noise rates are shown in Fig. 1-3. As shown in the figures, under varying noise rates, our method achieves relatively stable performance for both DI and EOd compared with other methods, which indicates that our method performs robustly under different noise rates.

### 4.3 Fairness under Noisy Sensitive Attributes and Noisy Labels

As discussed in Lemma 3, apart from sensitive attribute noise, our method can also be generalized to fairness under the exposure of label noise. Therefore, we also validate our method in the presence of both attribute and label noise, and results are shown in Table 4-6. While existing methods typically address one type of noise, our method is capable of handling both types of noise simultaneously, with better or comparable performance in terms of both fairness improvement and accuracy, and without requiring extra tools for noise rate estimation. This also validates our analysis in the previous section.

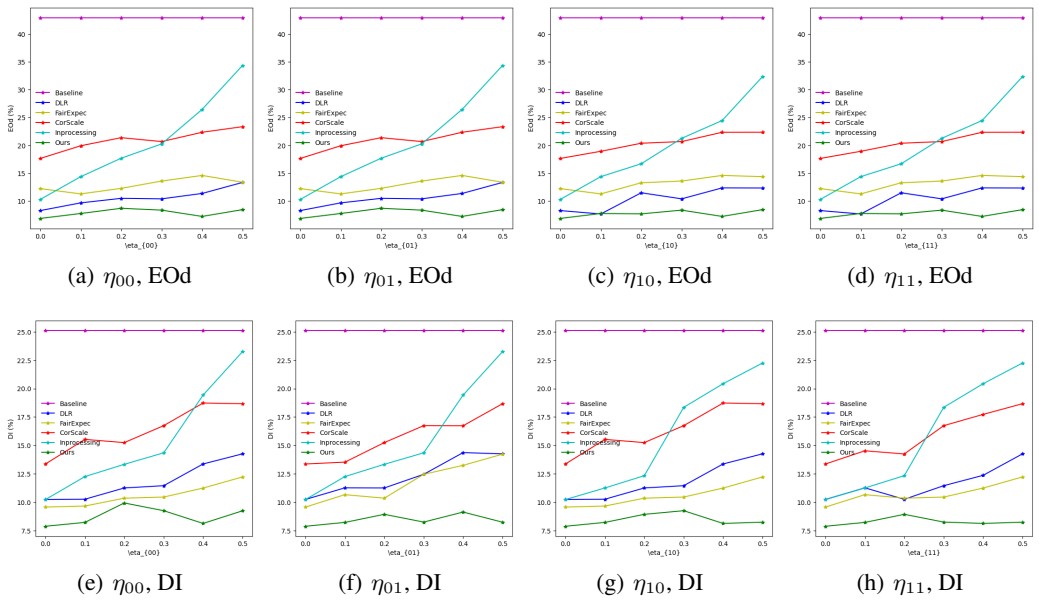

(a) $\eta_{00}$, EOd      (b) $\eta_{01}$, EOd      (c) $\eta_{10}$, EOd      (d) $\eta_{11}$, EOd

(e) $\eta_{00}$, DI      (f) $\eta_{01}$, DI      (g) $\eta_{10}$, DI      (h) $\eta_{11}$, DI

Figure 1: Change of EOd and DI as noise rates $\eta_{ya}$ vary on COMPAS dataset.

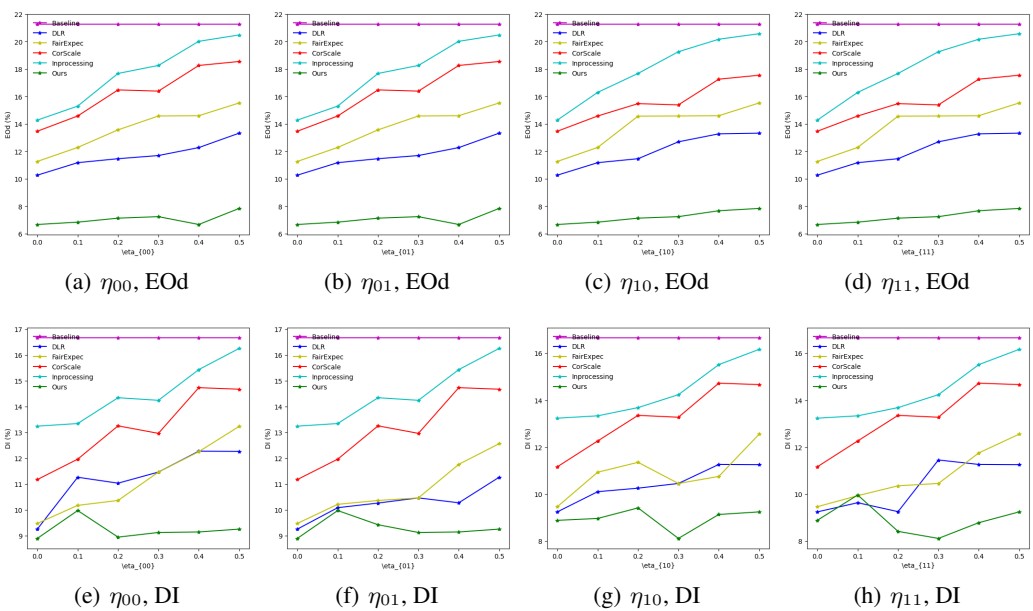

(a) $\eta_{00}$, EOd      (b) $\eta_{01}$, EOd      (c) $\eta_{10}$, EOd      (d) $\eta_{11}$, EOd

(e) $\eta_{00}$, DI      (f) $\eta_{01}$, DI      (g) $\eta_{10}$, DI      (h) $\eta_{11}$, DI

Figure 2: Change of EOd and DI as noise rates $\eta_{ya}$ vary on Adult dataset.

| Method | Accuracy | Disparate Impact | EOd |
|---|---|---|---|
| Baseline | 64.42±0.34% | 25.13±1.46% | 40.46±2.17% |
| Inprocessing (Wang et al., 2022) | 59.57±0.43% | 15.13±1.67% | 31.34±2.25% |
| DLR (Celis et al., 2021) | 58.57±0.92% | 12.25±1.67% | 16.45±2.17% |
| FairExpec (Mehrotra and Celis, 2021) | 59.23±1.24% | 9.43±1.47% | 11.64±1.67% |
| CorScale (Lamy et al., 2019) | 59.57±1.14% | 16.64±1.85% | 24.34±2.31% |
| SurrogateLoss (Wang et al., 2021) | 61.54±0.83% | 11.26±1.62% | 13.47±1.69% |
| Ours | 61.22±1.14% | 6.47±1.46% | 7.45±1.12% |

Table 4: Results on COMPAS dataset under label and sensitive attribute noise. The noise rates are set as $\eta_{00} = 0.2$, $\eta_{01} = 0.1$, $\eta_{10} = 0.3$, $\eta_{11} = 0.2$, $\beta_{00} = 0.35$, $\beta_{01} = 0.2$, $\beta_{10} = 0.15$, $\beta_{11} = 0.45$.

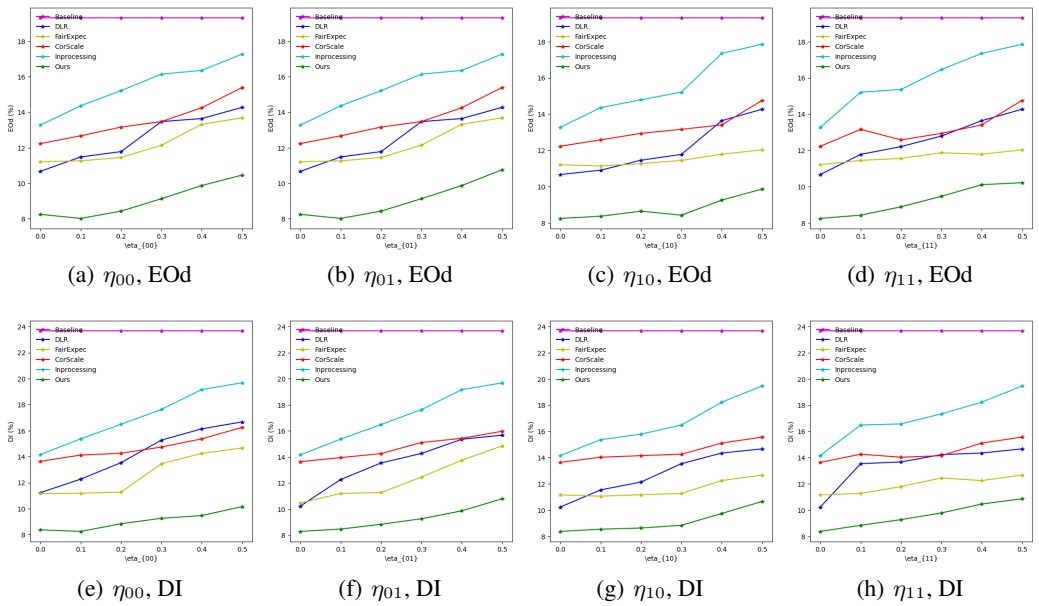

(a) $\eta_{00}$, EOd          (b) $\eta_{01}$, EOd          (c) $\eta_{10}$, EOd          (d) $\eta_{11}$, EOd

(e) $\eta_{00}$, DI          (f) $\eta_{01}$, DI          (g) $\eta_{10}$, DI          (h) $\eta_{11}$, DI

Figure 3: Change of EOd and DI as noise rates $\eta_{ya}$ vary on CelebA dataset.

| Method | Accuracy | Disparate Impact | EOd |
|---|---|---|---|
| Baseline | 81.54±0.85% | 16.85±1.65% | 21.75±1.42% |
| Inprocessing (Wang et al., 2022) | 77.46±0.58% | 14.27±1.48% | 16.63±1.25% |
| DLR (Celis et al., 2021) | 78.59±0.86% | 10.52±1.17% | 12.46±1.37% |
| FairExpec (Mehrotra and Celis, 2021) | 79.69±1.16% | 11.37±1.53% | 10.47±2.23% |
| CorScale (Lamy et al., 2019) | 78.76±1.24% | 12.66±1.83% | 15.43±1.76% |
| SurrogateLoss (Wang et al., 2021) | 79.14±1.56% | 11.56±1.35% | 12.67±1.52% |
| Ours | 80.27±0.67% | 8.56±1.67% | 7.47±1.85% |

Table 5: Results on Adult dataset under label and sensitive attribute noise. The noise rates are set as $\eta_{00} = 0.15$, $\eta_{01} = 0.1$, $\eta_{10} = 0.1$, $\eta_{11} = 0.3$, $\beta_{00} = 0.45$, $\beta_{01} = 0.3$, $\beta_{10} = 0.15$, $\beta_{11} = 0.35$.

| Method | Accuracy | Disparate Impact | EOd |
|---|---|---|---|
| Baseline | 87.23±0.69% | 21.27±1.83% | 19.34±1.28% |
| Inprocessing Wang et al. (2022) | 83.25±0.82% | 15.54±1.37% | 14.23±1.15% |
| DLR Celis et al. (2021) | 84.36±0.67% | 12.27±1.56% | 12.21±1.34% |
| FairExpec Mehrotra and Celis (2021) | 83.87±0.47% | 10.59±1.26% | 11.65±1.44% |
| CorScale Lamy et al. (2019) | 84.21±1.36% | 13.47±1.25% | 12.29±1.17% |
| SurrogateLoss Wang et al. (2021) | 85.23±0.69% | 12.37±1.64% | 11.16±1.43% |
| Ours | 85.11±0.69% | 9.74±1.28% | 8.78±1.27% |

Table 6: Results on CelebA dataset under label and sensitive attribute noise. The noise rates are set as $\eta_{00} = 0.1$, $\eta_{01} = 0.2$, $\eta_{10} = 0.3$, $\eta_{11} = 0.1$, $\beta_{00} = 0.25$, $\beta_{01} = 0.1$, $\beta_{10} = 0.15$, $\beta_{11} = 0.3$.

## 5 Conclusion

Fairness under noisy perturbation is an important yet less studied problem. In this paper, we formulate noisy perturbation as both group- and label-dependent, and we propose a fair representation learning framework based on normalizing flow to solve the problem without using extra tools for noise estimation. We prove theoretically the transferability of fairness between noisy and clean data under noisy sensitive attributes, and we show theoretically the connection between fairness measures of clean and noisy data under label noise. We validate from experiments that our method performs better or comparably in the improvement of fairness under both label noise and sensitive attributes noise generated under both static and varying noise rates, compared with state-of-the-art alternatives, with relatively small sacrifice in accuracy. Future directions include alternative methods for fair representation learning, and alternative formulations of noise perturbation.

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
