# OpenReview forum: "Fairness under Noise Perturbation: from the Perspective of Distribution Shift"
_NeurIPS.cc/2023/Conference — Submitted to NeurIPS 2023_

### Official Review · Reviewer_5tmy · 2023-06-20

**Soundness:** 3 good
**Presentation:** 2 fair
**Contribution:** 3 good
**Rating:** 4
**Confidence:** 4

**Summary:**

The authors introduce an innovative framework that enhances the fairness guarantees of a classifier in the presence of both sensitive attribute noise and label noise, considering them independently as well as in combination. Their approach incorporates theoretical guarantees and involves training a fair encoder to learn a novel data representation that ensures both fairness and accuracy. They demonstrate that by imposing bounded divergence between the noisy and clean distributions, fairness can be effectively transferred from one distribution to another. Notably, their method tackles the problem from a distribution shift perspective, eliminating the need for noise rate estimation typically required by conventional noise tolerant models.


**Strengths:**

- The introduction effectively substantiates all the claims made, including the contributions put forth by the authors. These assertions find validation through a thorough description of the methodology employed and the experiments conducted. The method section elaborates on the techniques and approaches considered, demonstrating how they align with the stated objectives. Furthermore, the experimental results provide empirical evidence that supports the claims made in the introduction.
- The problem addressed in the paper is well motivated. The authors provide a comprehensive and compelling rationale for the significance and relevance of the problem. They effectively highlight the real-world implications and potential consequences of the existing limitations in the field.
- The authors introduce an innovative alternative approach that effectively addresses the limitations of state-of-the-art (SOTA) methods. By identifying and highlighting the drawbacks of existing techniques (noise rate requirements), they demonstrate a clear understanding of the challenges at hand.
- The methodology is clearly explained and well-organized. The paper includes sub-sections that effectively delineate different aspects of the methodology, ensuring a coherent and structured presentation.
- The paper demonstrates commendable attention to reproducibility by providing thorough and detailed information regarding the experimental setup.
- For the experimental evaluation, the authors take into account various types of data, including both tabular and image data.
- The selection of datasets and the procedure employed to generate synthetic datasets align well with similar approaches found in the existing literature.
- The evaluation conducted in the paper is both sound and comprehensive. The authors meticulously consider various aspects to ensure a robust evaluation.


**Weaknesses:**

- (Section 4, Experiments) The authors put forth a proposition to tackle the challenge of ensuring fairness in the presence of noise from a distribution shift perspective. However, in the experimental section, they fail to compare their proposal with methods that specifically address distribution shift in a fairness-aware scenario. It is worth considering that these alternative methods may also yield promising results when handling noisy sensitive and label information. Including such comparisons would provide a more comprehensive understanding of the relative performance and effectiveness of the proposed approach within the context of fairness under distribution shift.
- (Section 4, Experiments) The paper presents theoretical bounds, but unfortunately, they are not evaluated empirically. While the theoretical analysis offers valuable insights and establishes the potential effectiveness of the proposed approach, the absence of empirical evaluations leaves room for uncertainty regarding its practical applicability. Empirical evaluations would have provided concrete evidence of the proposed method's performance and its ability to meet the expected bounds.
- (Section 2, Fairness metrics) The discussion of fairness metrics lacks a clear structure, and I would suggest that the authors differentiate between individual and group notions of fairness, providing distinct explanations for each. Additionally, it would be beneficial for the authors to acknowledge the emergence of mini-max fairness notions, which are gaining popularity in the field.
- (Section 2, Fairness-enhancing interventions)  While describing the pre-, in-, and post-processing methods, the authors primarily focus on specific techniques instead of providing an overview of the general framework. Consequently, it is not accurate to claim that all preprocessing methods aim to rectify the distribution of input features, nor is it true that all in-processing methods incorporate fairness enhancement as relaxed constraints. In reality, regarding the latter, there are variations where fairness is achieved through techniques such as fairness penalizations. While these cases are commonly encountered, it would be preferable for the authors to first describe the overarching objectives of the general workflows before delving into specific specifications. This approach would provide a clearer understanding of the broader goals before examining the specific techniques used. Moreover, the authors fail to explicitly state that their method constitutes an in-processing intervention.
- (Section 3.2) The authors initially discuss general distribution shift, but in line 167, they assert that they address covariate shift. It is important to note that these two types of shifts have distinct mathematical implications. Covariate shift specifically involves changes in p(x) between the source and target domains, while assuming that the functional form of p(y|x) remains unchanged. It would be beneficial for the authors to clarify which shift they are specifically addressing and how the mathematical characteristics of covariate shift come into play within their approach. Providing further clarity on this matter would help readers understand the specific focus and contributions of the proposed method in addressing the relevant shift.
- (Section 2, Fairness under distribution shift) In this section, the authors overlook several pertinent works, and some of the works mentioned are not even published. However, there exists a substantial body of literature that specifically addresses the challenge of ensuring fairness guarantees under distribution shift (for a comprehensive survey, the authors can refer to [1]). It is important to differentiate between methods that solely tackle distribution shift and those specifically designed for ensuring fairness under distribution shift. Furthermore, it is worth noting that different methods consider varying levels of data availability in the target domain, and not all of them assume the availability of (X, A) pairs [1]. For instance, the work [2] cited in that section assumes the target data is not available.
- (Section 2, Related works) The purpose of this section is to not only provide a description of the related works but also to establish the connection between them, elucidate the significance of these relationships, and highlight the novelty of the proposed work or its intended aim to address specific limitations. However, despite providing descriptions of various works, the authors do not explicitly specify the precise position of their work within the broader landscape.
- Notation issues: After defining the notation at the beginning of Section 3, and Section 3.1, the authors employ symbols that have not been defined, such as, A in line 137, or $\mathcal{L}_{cls}$  in Eq (6). Regarding the latter, there is no specification regarding its meaning nor its mathematical form.
- The paper contains several typos: line 67 after more there is a full stop, line 129 after of there should be an 'a', line 217 let should be in uppercase.

[1] Barrainkua, A., Gordaliza, P., Lozano, J. A., & Quadrianto, N. (2022). A Survey on Preserving Fairness Guarantees in Changing Environments. arXiv preprint arXiv:2211.07530.

[2] Rezaei, A., Fathony, R., Memarrast, O., & Ziebart, B. (2020, April). Fairness for robust log loss classification. In Proceedings of the AAAI Conference on Artificial Intelligence (Vol. 34, No. 04, pp. 5511-5518).


**Questions:**

- Theoretical results in the paper are presented in two different contexts: some are with respect to P, while others are with respect to Q. However, the relationship between these two contexts is not clearly established or explained. How are they related?
- The framework presented in this study focuses on a binary sensitive attribute and a binary label, which may limit its applicability in more complex scenarios commonly encountered in various applications. For example, many SOTA noise tolerant approaches cited in the work can handle those situations. But can this framework be extended to support multiclass Y or multivalue S? Moreover, can it effectively handle scenarios involving multi-dimensional S? Further clarification is needed to assess the flexibility and scalability of the proposed framework in handling these additional complexities.
- Does your approach primarily consider general distribution shift or does it specifically focus on covariate shift?
- Why have you only chosen DI and EOd as fairness metrics? Can it be extended to other statistical notions?




**Limitations:**

The authors do not thoroughly discuss the limitations of their method, which is an important aspect to consider. Taking inspiration from the questions raised concerning the shift type and potential implications beyond binary Y and binary S could be valuable in addressing the limitations and further refining their approach.

---

> ### Author Rebuttal · Authors · 2023-08-10
>
> We thank the reviewer for the comment. We'll fix the typos and include the suggested reference and discussions in final paper. For **Weakness 1 (W1): Comparison with fairness under distribution shift**, **Question 1 (Q1): Connection between $P$ and $Q$**, **Q2: Non-binary setting** and **Limitations**, please refer to the global rebuttal.
>
> **[W2: Verification of theoretical bounds]** Thanks for the suggestion. We include results on empirical verification of theoretical bounds as stated in Theorem 1. For simplicity of expression, we denote $\text{EOd}^{\text{upper}} := \hat{\text{EOd}}+\frac{\eta_{00}+\eta_{01}}{1-\eta_{00}-\eta_{01}} \sqrt{\epsilon_0}+\frac{\eta_{10}+\eta_{11}}{1-\eta_{10}-\eta_{11}} \sqrt{\epsilon_1}$ as the upper-bound of $\text{EOd}$ stated in Theorem 1. Results are shown as follows:
>
> **Table 1: Empirical verification of Theorem 1 on COMPAS, Adult and CelebA dataset regarding our method. The noise rates are set the same as in Tab. 1-3 of our paper.**
> Dataset|$\hat{\text{EOd}}$|EOd|$\text{EOd}^{\text{upper}}$
> -|-|-|-
> COMPAS|0.06|0.09|0.13
> Adult|0.04|0.07|0.10
> CelebA|0.04|0.08|0.11
>
> This shows that the upper-bound $\text{EOd}^{\text{upper}}$ in Theorem 1 serves as a good approximation of $\text{EOd}$, which thereby verifies the practical applicability of Theorem 1.
>
> **[W3: Fairness metrics]** Thanks for the kind suggestion. We'll include more discussion and explanation on individual and group fairness notions, and clarify that our method focuses on the group fairness notion. Also, we will discuss mini-max fairness notion in group fairness in final paper attentively, so as to better distinguish between different notions.
>
> **[W4, W6, W7, Q3: Fairness-enhancing interventions and related works]** Thanks for the detailed suggestion. As the reviewer's questions are concerned about connection with related works, position of our work and discussions regarding related works on fairness, we combine the responses into one paragraph for a more complete response. We'll carefully revise the phrasing for the general framework of pre-, in-, and post-processing methods in final paper, as well as the subsection on fairness under distribution shift to include more recent works on this topic and to provide a more precise description on different assumptions in data availability in the target domain. Besides, as suggested by the reviewer, our method is an in-processing intervention.
>
> Our method are primarily focused on solving the fairness under noise perturbation problem from the perspective of covariate shift. Compared with previous work on fairness under noise perturbation, our method does not require estimation of noise rate, which reduces both the computational complexity and the potential deviation in the estimation. Compared with work on fairness under distribution shift that does not require knowledge on target data (for instance, work [1] formulates the problem as a mini-max game), our method makes several distinct contributions:
> - 1) we provide provable fairness guarantee under varying noise rates, due to the invertibility of normalizing flows, as stated in Theorem 1 of our paper;
> - 2) since we are primarily concerned about the very type of distribution shift induced by sensitive attribute noise (i.e., we intend to solve fairness under noise perturbation from the perspective of distribution shift), rather than the worst-case approximation of shift [1], our method achieves better performance in terms of fairness and minimum sacrifice in accuracy under different datasets under the very type of covariate shift induced by noisy sensitive attribute $a$. We validate this by experimental results in Tab. 1 of global response, where we compare with distribution-agnostic method [1], as suggested by the reviewer, as well as method that requires knowledge on target domain [2].
> - 3) we validate in subsection 4.3 of our paper that our method also works under label shift, and our method can be applied under simultaneous exposure of both label shift and covariate shift induced by sensitive attribute noise.
>
> **[W5: Connection with fairness under distribution shift]** We are sorry for the confusion. Our discussion regarding distribution shift contains two different parts: covariate shift, which we use to model fairness under sensitive attribute noise, and label shift, which is directly related to the discussion regarding label noise in Lemma 3. We clarify that we do not intend to address fairness under distribution shift problem in general; rather, we are primarily focused on the connection between covariate shift and fairness under sensitive attribute noise, and we try to solve the problem of fairness under sensitive attribute noise from the perspective of fairness under covariate shift.
>
> **[W8: Notation issues]** Thanks for the suggestion. $A$ in Line 137 refers to the random variable that corresponds to sensitive attribute, as defined in Line 124 in our paper. $\mathcal{L}\_{cls}$ refers to classification loss. Under binary setting, we choose $\mathcal{L}\_{cls}$ to be cross-entropy loss.
>
> **[Q4: Extension to other metrics]** We would like to clarify that our primary focus in our training objective is EOd, and we report results of DI as it is a widely adopted metric for fairness. We include results on worst-group accuracy that corresponds to the mini-max notions in fairness in Tab. 3 in the attached PDF in the global rebuttal. Compared with baseline method, our method also improves worst-group accuracy, in regard of mini-max fairness. We'll include full results in final paper.
>
>
> [1] Rezaei, Ashkan, et al. "Fairness for robust log loss classification." Proceedings of the AAAI Conference on Artificial Intelligence. Vol. 34. No. 04. 2020.
>
> [2] An, Bang, et al. "Transferring fairness under distribution shifts via fair consistency regularization." Advances in Neural Information Processing Systems 35 (2022): 32582-32597.

---

> > ### Comment · Reviewer_5tmy · 2023-08-20
> >
> > Thank you very much for your response.
> > Regarding W2 the upper bound is as close to the true value than the $\hat{EOd}$ is, thus why is it a better aporoximation?
> > Besides, I still believe that a deeper evaluation needs to be carried out, including the methods that address distribution shift explicitly.
> > Therefore, I have decided to keep the rating.

---

> > > ### Author Response · Authors · 2023-08-20
> > > **Follow-up to Reviewer 5tmy**
> > >
> > > Thank you for your response.
> > >
> > > - 1) We would like to clarify that we do not suggest $\text{EOd}^{\text{upper}}$ a better approximation of $\text{EOd}$ than $\hat{\text{EOd}}$. Instead, our results in **[W2: Verification of theoretical bounds]** show that $\text{EOd}^{\text{upper}}$ serves as a good approximation of $\text{EOd}$. Moreover, as $\text{EOd}^{\text{upper}}$ serves as an upper-bound of $\text{EOd}$, by minimizing $\text{EOd}^{\text{upper}}$ we are minimizing the upper-bound of $\text{EOd}$, instead of the lower-bound of $\text{EOd}$ by $\hat{\text{EOd}}$.
> > >
> > > - 2) Per the reviewer's suggestion, we show comparison with methods that address fairness under distribution shift explicitly [1,2] in Tab. 1 of global rebuttal. The results show that our method achieves better improvement in fairness with better or comparable performance in accuracy. We 'll include the evaluation results and discussions in final paper.
> > >
> > > [1] Rezaei, Ashkan, et al. "Fairness for robust log loss classification." Proceedings of the AAAI Conference on Artificial Intelligence. Vol. 34. No. 04. 2020.
> > >
> > > [2] An, Bang, et al. "Transferring fairness under distribution shifts via fair consistency regularization." Advances in Neural Information Processing Systems 35 (2022): 32582-32597.

---

### Official Review · Reviewer_fq6s · 2023-07-07

**Soundness:** 2 fair
**Presentation:** 2 fair
**Contribution:** 2 fair
**Rating:** 5
**Confidence:** 4

**Summary:**

The paper aims to improve the performances of fair training when the group attributes or labels in the training data have noisy information. The paper views the noisy training data problem as a kind of distribution shift, where the training data is noisy and the test data is clean. To address this issue, the paper proposes a fair representation learning method to reduce the impact of distribution differences. The paper also provides some theoretical analyses to show the relationship between the group fairness results and noisy training data. In the experiment, the paper uses three datasets and compares with several baselines to show the performance gains of the proposed method.


**Strengths:**

S1. The paper solves an important research problem, preserving the performances of fair training under the noisy training data. The paper views this problem as a distribution shift issue.

S2. The paper gives some theoretical analyses on the relationship between the group fairness and noisy data.

S3. The proposed algorithm empirically shows better fairness and accuracy performances compared to the baselines.


**Weaknesses:**

W1. Many important details are missing in the proposed fair representation learning.
- In Section 3.3, the final training objective in Eq. (6) has many unexplained important details. For example, what is L_cls, and how are the input arguments (e.g., g_00, h) used in L_cls? Also, it seems the lambda values are the tuning knobs, but there is no explanation on why the loss terms should be connected by two lambda values. Including these details, a clearer rationale for design choices is needed.

W2. In experiments, the proposed algorithm is not clearly analyzed. For example, it would be much better if the paper explains the following.
- How the lambda values in Equation 6 affect the training performances
- The computational complexity of the proposed algorithm

W3. Although this work is highly related to the studies on fairness under data distribution shifts, there are no clear comparisons with them. In experiments, all the baselines are from the noisy training literature. Since many algorithms for fair training under distribution shifts have been recently proposed, it would be better to compare with them empirically or at least to be clearly discussed.


**Questions:**

All questions are included in the above weakness section.

**Limitations:**

The paper did not discuss the limitations and possible negative societal impacts. As the limitations, this work may discuss which types of data noises cannot be handled by the proposed algorithm.

---

> ### Author Rebuttal · Authors · 2023-08-10
>
> We thank the reviewer for the comment. For **Weakness 3: Comparison with fairness under distribution shift** and **Limitations**, please refer to the '**Comparison with fairness under distribution shift**'  and '**Limitations**' parts in global rebuttal.
>
> **[Weakness 1 (W1): Details of training objective]** We are sorry for the confusion. Under binary classification, $\mathcal{L}\_{cls}$ can be chosen as cross-entropy loss, i.e., $\mathcal{L}\_{cls}=-\frac{1}{N}\sum_{i=1}^N [y_i\log(h(g_{y_i a_i}(x_i)))+(1-y_i)\log(1-h(g_{y_i a_i}(x_i)))]$. For classification, our framework involves two parts as defined in Line 183-184, a bijective encoder $g$, which maps the input feature to a latent representation, and the classification head $h$, which maps the latent representation to the predicted soft label. We use two different hyperparameters for $\mathcal{L}\_{0}$ and $\mathcal{L}\_{1}$, as we formulate sensitive attribute noise to be both group- and class-dependent. This leads to different coefficients in terms of disparities in TPR and disparities TNR under clean data, compared with those under noisy data:
> $$
> |\hat{\text{TPR}\_0} - \hat{\text{TPR}\_1}| = |(1-\eta\_{10})\text{TPR}\_0 + \eta\_{10}\text{TPR}\_1 - (1-\eta\_{11})\text{TPR}\_1 - \eta\_{11}\text{TPR}\_0| = (1-\eta\_{10}-\eta\_{11})\text{DTPR},
> $$
> $$
> |\hat{\text{TNR}\_0} - \hat{\text{TNR}\_1}| = |(1-\eta\_{00})\text{TNR}\_0 + \eta\_{00}\text{TNR}\_1 - (1-\eta\_{01})\text{TNR}\_1 - \eta\_{01}\text{TNR}\_0| = (1-\eta\_{00}-\eta\_{01})\text{DTNR}.
> $$
>
> Therefore, the hyperparameters $\lambda\_0$ and $\lambda\_1$ for fairness regularization in Eq. 6 are not necessarily identical, so as to align with the possible difference in noise rates, and connecting $\mathcal{L}\_{0}$ and $\mathcal{L}\_{1}$ with different hyperparameters brings us more flexibility in the presence of sensitive attribute noise. We'll include more details regarding the training objective in final paper to provide a clearer rationale for the design choices.
>
> **[W2: Effect of $\lambda$ values]** Thanks for the suggestion. We include more results on the trade-off between fairness and accuracy as $\lambda_0$ and $\lambda_1$ vary under different noise ratios in Fig. 1 of global rebuttal. As shown in the figure, under different noise rates, our method shows similar fairness-utility trade-off, where fairness gradually improves as $\lambda_0$ and $\lambda_1$ increases, and the fairness improvement becomes smaller as $\lambda_0$ and $\lambda_1$ increase.
>
> **[W2: Computational complexity]** The update of our normalizing flow framework involves computing the determinant for each layer at each training iteration. Generally, the time complexty of this operation is $\mathcal{O}(n^3)$, where $n$ is the input feature dimension [1].
>
> **[Potential societal impacts]** One possible societal impact is that, while our method deals with fairness under noise perturbation, we still need access to noisy sensitive information during training. There could lead to the concern privacy issues, even though with a reduced risk due to noise perturbation. We'll add the discussion in final paper.
>
> [1] Keller, Thomas A., et al. "Self normalizing flows." International Conference on Machine Learning. PMLR, 2021.

---

> > ### Comment · Reviewer_fq6s · 2023-08-18
> > **Thank you for the response.**
> >
> > I appreciate the author's response.
> >
> > After reading the response, many of my concerns are resolved. I thus updated my score.
> >
> > There is one follow-up question regarding computational complexity. It seems the O(n^3) time complexity can be a notable bottleneck in large-scale settings. It would be helpful if the revised version of the paper could give any suggestions on handling such scenarios.

---

> > > ### Author Response · Authors · 2023-08-18
> > > **Thank you**
> > >
> > > Thank you for taking the time and effort to review our work and we appreciate your recognition of our work. For the computational complexity, we have $n$ the input feature dimension, and different approaches on flow-based model have been proposed to reduce the computational complexity [1,2,3]. We 'll include more discussions on this topic in final version.
> > >
> > > [1] Hoogeboom, Emiel, Rianne Van Den Berg, and Max Welling. "Emerging convolutions for generative normalizing flows." International conference on machine learning. PMLR, 2019.
> > >
> > > [2] Keller, Thomas A., et al. "Self normalizing flows." International Conference on Machine Learning. PMLR, 2021.
> > >
> > > [3] Caterini, Anthony L., et al. "Rectangular flows for manifold learning." Advances in Neural Information Processing Systems 34 (2021): 30228-30241.

---

### Official Review · Reviewer_Dpoc · 2023-07-09

**Soundness:** 4 excellent
**Presentation:** 4 excellent
**Contribution:** 3 good
**Rating:** 6
**Confidence:** 4

**Summary:**

The paper studies the fairness problem under noise perturbation on both label and sensitive attributes. In particular, it considers such a problem from the perspective of distribution shift and uses the normalizing flow framework to analyze the problem. Empirically, the proposed methods achieve the best utility and fairness trade-offs under different settings of noise perturbation.

**Strengths:**

1. The paper presents a method for learning fair representation when there is noise on both sensitive attributes and labels. The method is straightforward and empirically shown to be effective.
2. The theoretical analysis is sound
3. Compared to the previous work, this work considers both label and sensitive attribute noise without directly estimation the noise parameter, which is more practical in real-world applications.


**Weaknesses:**

I do not find any obvious weaknesses in the paper. But there are minor points that the author could further improve their paper.
1. The assumption of invertible function in the fair normalizing flow methods might be strong. For example, In ResNet, the default activation function is ReLU, which is not invertible. The authors might need to provide more justification for this.
2. Discussion of limitations. The paper could be improved if there is a discussion of the limitations.


**Questions:**

1. Is there a utility-fairness trade-off tuning parameters in your methods? If yes, how does the utility-fairness trade-offs change given different noise rate?

**Limitations:**

The authors do not discuss the limitations of the work, which is highly suggested.

---

> ### Author Rebuttal · Authors · 2023-08-10
>
> We thank the reviewer for the comment. For **Weakness 2: Discussion of limitations**, please refer to the '**Limitations**' part of global rebuttal.
>
> **[Weakness 1 (W1): Assumption of invertible function in normalizing flow]** The invertibility is a basic formulation in normalizing flow methods [1,2,3]. We choose normalizing flow framework as it not only provides promising performance in fairness [4], but it also enables us to compute the exact likelihood in the latent space, which provides us provable fairness guarantee in terms of the statistical divergence between latent representations of different subgroups. Moreover, while certain activation functions including ReLU are not invertible and can not be applied to normalizing flow, recent work on normalizing flow has shown promising performance compared with state-of-the-art methods [5,6,7].
>
> **[Question 1: Utility-fairness trade-off]** Thanks for the suggestion. We include results on fairness-utility trade-off in Fig. 1 of global rebuttal. Under different noise rates, our method shows similar fairness-utility trade-off, where fairness gradually improves as $\lambda\_0$ and $\lambda\_1$ increase, and the fairness improvement becomes smaller as $\lambda\_0$ and $\lambda\_1$ increase.
>
> [1] Kobyzev, Ivan, Simon JD Prince, and Marcus A. Brubaker. "Normalizing flows: An introduction and review of current methods." IEEE transactions on pattern analysis and machine intelligence 43.11 (2020): 3964-3979.
>
> [2] Papamakarios, George, et al. "Normalizing flows for probabilistic modeling and inference." The Journal of Machine Learning Research 22.1 (2021): 2617-2680.
>
> [3] Rezende, Danilo, and Shakir Mohamed. "Variational inference with normalizing flows." International conference on machine learning. PMLR, 2015.
>
> [4] Balunović, Mislav, Anian Ruoss, and Martin Vechev. "Fair normalizing flows." arXiv preprint arXiv:2106.05937 (2021).
>
> [5] Izmailov, Pavel, et al. "Semi-supervised learning with normalizing flows." International Conference on Machine Learning. PMLR, 2020.
>
> [6] Mackowiak, Radek, et al. "Generative classifiers as a basis for trustworthy image classification." Proceedings of the IEEE/CVF Conference on Computer Vision and Pattern Recognition. 2021.
>
> [7] Wang, Tianchun, et al. "GC-Flow: A Graph-Based Flow Network for Effective Clustering." arXiv preprint arXiv:2305.17284 (2023).

---

### Official Review · Reviewer_Vz2U · 2023-07-10

**Soundness:** 3 good
**Presentation:** 2 fair
**Contribution:** 3 good
**Rating:** 6
**Confidence:** 3

**Summary:**

This work studies noise tolerance of fairness from the perspective of subpopulation/subgroup shift, by considering the perturbation of the sensitive attributes as well as the labels _without_ the need for noise-rate estimation - by considering the noisy distribution as the source and clean distribution as the target. This leads to a "covariate" shift between the source and the target distributions, with the shift being a consequence of the noise. The work then proposes a fair representation learning method for fairness under noisy attributes based on normalizing flows, and presents a theoretical result showing that this method minimizes the upper-bound of the clean equalized odds. Thorough empirical evaluation is presented for both static and varying noise rates, showing the efficacy of the proposed method.

**Strengths:**

This work addresses an important setting of achieving fairness when both the label $y$ and sensitive attribute $a$ can be noisy - as traditional metrics can be biased under noisy data. The theoretical analysis presents a comprehensive study of fairness transfer between clean and noisy data and supports the choice of the minimizer in the proposed method. The empirical analysis is thorough - the section 4.2.2 is especially interesting as it considers both static and dynamic noise rates.


**Weaknesses:**

The following points should be considered:
1. The loss function (in Eq. 6) focuses on a binary valued $a$ and $y$. How does this methodology extend to the more general case either/both can be multi-valued? Is it straightforward?
2. Is there any more intuition on leveraging normalized flows for this setting?




**Questions:**

Please see Weakness section.

A minor suggestion:
1. Consider using a bigger text size for the plots as they are unreadable.
2. Consider using more formal English: e.g. line 67 "What's more.."

**Limitations:**

No limitations of this method are mentioned, and it would be nice if any potential drawbacks can be discussed.

---

> ### Author Rebuttal · Authors · 2023-08-10
>
> We thank the reviewer for the comment. We'll refine the writing and adjust the text size for plots in final paper. For **Weakness 1 (W1): Multi-valued version of loss function** and **Limitations**, please refer to the '**Non-binary setting**' and '**Limitation**' parts of global rebuttal.
>
> **[W2: Using normalizing flow]** As discussed in section 3.2, our main goal is to minimize the divergence between distributions of predicted soft labels in different subgroups by minimizing the divergence between distributions of data in different subgroups. Normalizing flow enable us to compute the exact likelihood in the latent space [1,2,3], which can be used to provide provable fairness guarantee in terms of the statistical divergence between latent representations of different subgroups.
>
> [1] Dinh, Laurent, Jascha Sohl-Dickstein, and Samy Bengio. "Density estimation using Real NVP." International Conference on Learning Representations. 2016.
>
> [2] Rezende, Danilo, and Shakir Mohamed. "Variational inference with normalizing flows." International conference on machine learning. PMLR, 2015.
>
> [3] Kobyzev, Ivan, Simon JD Prince, and Marcus A. Brubaker. "Normalizing flows: An introduction and review of current methods." IEEE transactions on pattern analysis and machine intelligence 43.11 (2020): 3964-3979.

---

> > ### Comment · Reviewer_Vz2U · 2023-08-20
> >
> > Thank you Authors for clarifying, I acknowledge that I have gone through the rebuttal. I would recommend adding the analysis on multi-valued $a$ and $y$ in the revised paper.

---

> > > ### Author Response · Authors · 2023-08-20
> > > **Thank you**
> > >
> > > Dear Reviewer,
> > >
> > > Thank you for taking the time and effort to review our work and we appreciate your recognition of our work. We 'll include the analysis on multi-valued $a$ and $y$ in the revised paper accordingly.
> > >
> > > Best, Authors

---

### Official Review · Reviewer_vtBi · 2023-07-20

**Soundness:** 3 good
**Presentation:** 2 fair
**Contribution:** 3 good
**Rating:** 5
**Confidence:** 1

**Summary:**

This paper targets the problem of ensuring fairness with noises on either sensitive attributes or labels. Specifically, this paper models the noisy data training set and clean test set as a distribution shift and proposes a regularization term to improve the fairness of classifiers.
The theoretical analysis indicates that the classifier trained by the proposed framework on the noise data can be bounded when evaluating on the clean data.

**Strengths:**

1. The problem of ensuring fairness when training on noisy data is an important problem.

2. The whole framework makes sense.

3. The experimental results show the advantage of the proposed approach.

**Weaknesses:**

The writing is not very friendly to readers without a background in this specific fairness problem. Please check my questions below.

**Questions:**

1. One contribution claimed in this paper is that the proposed framework does not require noise rate estimation. However, if I understand correctly, \lambda in the objective function (Eq 6) is a function of $\eta$ and $\hat{\alpha}$ defined in Lemma 1. While the $\eta$ indicates the noise rate in a specific subgroup, I am not convinced that not requiring the noise rate estimation is a contribution. It seems like the proposed framework needs knowledge of the noise rate for each subgroup in advance.

2. As I am not an expert in the fairness issue under the noise data setting, some equations are not very straightforward to me, such as the equation under Line 138 and Eqs 1 and 2. It would be better to give more explanations on those equations. Especially, Eq 2 is important for the conclusion described in Line 177.

3. The proposed framework models the noisy training data and clean testing data from the perspective of distribution shift. The high level idea makes sense to me. However, I do not follow the description between lines 173-174. Why does minimizing the divergence between \hat{P}_{ya} and \hat{P}_{ya'} lead to the minimization of  the divergence between $Q_{ya}$ and $\hat{Q}_{ya}$?

**Limitations:**

Please check my questions above.

---

> ### Author Rebuttal · Authors · 2023-08-10
>
> We thank the reviewer for the comment. For **Question 3: Connection between KL-divergence**, please refer to the '**Connection between $P$ and $Q$**' part of global rebuttal.
>
> **[Question 1 (Q1): Requirement of noise rate estimation]** We are sorry about the confusion. $\lambda_0$ and $\lambda_1$ in the objective function (Eq. 6) are hyperparameters that control the trade-off between classification loss and fairness regularization, and are **different** from the $\lambda_a$ defined in Lemma 1. Tuning hyperparameters $\lambda_0$ and $\lambda_1$ in the objective function (Eq. 6) does not require noise rate estimation. We'll revise the notions in final paper to avoid repeated use.
>
> **[Q2: Explanation of line 138, Eq. 1 and 2]** We are sorry for the confusion and we'll include more explanation regarding the formulation in final paper. For Eq. 1, our formulation of $\eta_{ya}:= p\left[A \neq \hat{A}|Y=y,\hat{A}=a\right]$ assumes random flips within different sensitive subgroups, and flips between different subgroups are independent. Therefore, we can decompose the distribution of data $\hat{P}\_{ya}$ under noisy sensitive group $\\{Y=y,\hat{A}=a\\}$ as compositions of distributions under clean sensitive groups $P_{ya}$ and $P\_{ya'}$:
> $$
> \hat{P}\_{ya} = \eta_{ya} P_{ya'} + (1-\eta_{ya}) P_{ya},
> $$
> where $\eta_a P_{ya'}$ resembles samples from the clean subgroup $\\{Y=y,A=a'\\}$ whose sensitive information are flipped, and $(1-\eta_a) P_{ya}$ resembles samples from the clean subgroup $\\{Y=y,A=a\\}$ whose sensitive information remain unchanged (i.e., not affected by noise). Therefore we have the following equation set regarding $P_{ya}$, $P_{ya'}$, $\hat{P}\_{ya}$ and $\hat{P}\_{ya}$:
> \begin{align*}
> \hat{P}\_{ya} &= \eta_{ya} P_{ya'} + (1-\eta_{ya}) P_{ya}, \\\\
> \hat{P}\_{ya'} &= \eta_{ya'} P_{ya} + (1-\eta_{ya'}) P_{ya'}.
> \end{align*}
>
> By solving the equation set above we are able to obtain the expression of $P\_{ya}$ in terms of $\hat{P}\_{ya}$ and $\hat{P}\_{ya'}$ as in Eq. 2.
>
> Line 138 follows similar formulation, except that the noise rate $\eta_a := p\left[A \neq \hat{A}|\hat{A}=a\right]$ now becomes identical within each sensitive group, and we have $\hat{P}\_{ya} = \eta_a P\_{ya'} + (1-\eta_a) P\_{ya}$ and $\hat{P}\_{a} = \eta\_a P\_{a'} + (1-\eta\_a) P\_{a}$ under such formulation. The fairness measures under $\eta_a$ follows by substituting $\hat{P}\_{a}$ and $\hat{P}\_{ya}$ with the corresponding clean distributions:
> \begin{align*}
> \hat{\text{DI}} &= \int\_{0.5}^{1}|\hat{P}\_{0}-\hat{P}\_{1}| = \int\_{0.5}^{1}|(\eta\_0 P\_{1} + (1-\eta\_0) P\_{0}) - (\eta\_1 P\_{0} + (1-\eta\_1) P\_{1})| \\\\
> &= (1-\eta\_0-\eta\_1) \int\_{0.5}^{1}|{P}\_{0}-{P}\_{1}| = (1-\eta\_0-\eta\_1)\text{DI},
> \end{align*}
>
> \begin{align*}
> \hat{\text{EOd}} = & \int\_{0.5}^{1}|\hat{P}\_{10}-\hat{P}\_{11}| + \int_{0.5}^{1}|\hat{P}\_{00}-\hat{P}\_{01}|
> \\\\
> = &\int\_{0.5}^{1}|(\eta\_0 P\_{11} + (1-\eta\_0) P\_{10}) - (\eta\_1 P\_{10}+ (1-\eta\_1) P\_{11})|
> \\\\
> &+ \int\_{0.5}^{1}|(\eta\_0 P\_{01} + (1-\eta\_0) P\_{00}) - (\eta\_1 P\_{00} + (1-\eta\_1) P\_{01})|= (1-\eta\_0-\eta\_1)\text{EOd}.
> \end{align*}

---

> > ### Comment · Reviewer_vtBi · 2023-08-14
> >
> > Thanks, authors, for the clarification.  I suggest authors include more details in the next version. Meanwhile, as $a$ could be either 0 or 1, it would be better to use other notations to represent the hyper-parameters in Eq 6.

---

> > > ### Author Response · Authors · 2023-08-15
> > > **Thank you**
> > >
> > > Dear Reviewer,
> > >
> > > Thank you for taking the time and effort to review our work. We sincerely appreciate your constructive feedback and we will revise our final paper accordingly.
> > >
> > > Best, Authors

---

### Author Rebuttal · Authors · 2023-08-10

We thank the reviewer for the detailed comment and we would like address issues that are widely concerned by reviewers in this global rebuttal.

**[Comparison with fairness under distribution shift]** We compare our method with two state-of-the-art methods [1,2] on fairness under distribution shift under the same setting as the experiments in our paper. Results are shown in Tab. 1 in the attached PDF. Compared with methods on fairness under distribution shift, our method perform better or competitively in terms of fairness and accuracy, which validates the effectiveness of our method. We'll include the results in final paper.

**[Connection between $P$ and $Q$]** Let $f$ be function of classifier (namely the composition of bijective encoders $g\_{ya}$ and classification head $h$ for our method, as defined in Line 183-184), we have $f$ pushes $\hat{P}\_{ya'}$ and $\hat{P}\_{ya}$ forward to $\hat{Q}\_{ya'}$ and $\hat{Q}\_{ya}$. Therefore, by data processing inequality [3] we have
$$
D\_{KL}(\hat{P}\_{ya'}||\hat{P}\_{ya}) \ge D\_{KL}(\hat{Q}\_{ya'}||\hat{Q}\_{ya}),
$$
which suggests that by minimizing the divergence between $\hat{P}\_{ya'}$ and $\hat{P}\_{ya}$ we are able to minimize the upper-bound of divergence between $\hat{Q}\_{ya'}$ and $\hat{Q}\_{ya}$. Similar to Eq. 3 in our paper, we have the following relationship regarding $\hat{Q}\_{ya}$ and $Q\_{ya}$ by the formulation of noise perturbation:
$$
D\_{K L}\left(\hat{Q}\_{ya}||Q\_{ya}\right)=\int \hat{Q}\_{ya} \log \frac{\hat{Q}\_{ya}}{Q\_{ya}}=-\int \hat{Q}\_{ya} \log \left[\frac{1-\eta\_{ya^{\prime}}}{1-\eta_{ya}-\eta_{ya^{\prime}}}-\frac{\eta_{ya} \frac{\hat{Q}\_{ya^{\prime}}}{\hat{Q}\_{ya}}}{1-\eta\_{ya}-\eta\_{ya^{\prime}}}\right],
$$
which suggests that by minimizing the divergence between $\hat{Q}\_{ya'}$ and $\hat{Q}\_{ya}$ we also minimizes the divergence between $\hat{Q}\_{ya}$ and ${Q}\_{ya}$. Specifically, since $D_{KL} \ge 0$, when $D\_{KL}(\hat{P}\_{ya'}||\hat{P}\_{ya})=0$, we also have $D\_{KL}(\hat{Q}\_{ya}||Q\_{ya})=0$.

**[Non-binary setting]** Our methodology can be readily generalized to multi-classes (i.e., multi-valued $Y$) and multi-valued sensitive attributes (i.e., multi-valued $A$). Similar to the binary setting, we have different subgroups specified by label and sensitive information. Therefore we apply bijective encoder $g_{ya}: \mathbb{R}^n \rightarrow \mathbb{R}^d, \mathbf{x} \mapsto \mathbf{z}$ to map sample $\mathbf{x}$ in the corresponding subgroup $\\{Y=y,A=a\\}$ to latent representation $\mathbf{z}$, and adjust the classification head $h: \mathbb{R}^d \rightarrow \\{0,1\\}^c, \mathbf{z} \mapsto \mathbf{y}$ accordingly to fit with the class number $c$. The corresponding network contains several bijective encoders $g_{ya}$ and one universal classification head $h$. The $\mathcal{L}\_0$ and $\mathcal{L}\_1$ terms in Eq. 6 in our paper now becomes pair-wise symmetrized divergence between subgroups, i.e., $\mathcal{L}\_y = \sum_{a} \sum_{a'\neq a} D\_{KL}(P\_{z\_{ya}},P\_{z\_{ya'}}) + D_{KL}(P\_{z\_{ya'}},P\_{z\_{ya}})$ and $\mathcal{L}\_{cls}$ in Eq. 6 in our paper becomes negative log likelihood loss for multi-class classification.

We include results on COMPAS dataset under **multi-dimensional $A$** in Tab. 2 in the attached PDF to empirically verify the generalization. The sensitive attribute is chosen as $\text{race} \times \text{sex}$ the vector $[a_{\text{race}},a_{\text{sex}}]$. Under multi-dimensional $A$, our method still shows significant improvement in terms of fairness compared with baseline with relatively small sacrifice in accuracy.

We also include results on CRIME dataset [4] to validate the performance of our method under **multi-class classification**. The task is to predict the number of violent crimes per $10^5$ population, and we divide the numbers into $K = 4$ classes based on equidistant quantiles. The sensitive attribute is chosen as ethnicity. As shown in Tab. 2 in the attached PDF, our method achieves remarkable improvement in fairness with relatively small decrease in classification accuracy under multi-class scenarios. We'll include full results in final paper.

**[Limitations]** One potential limitation is the formulation of noise perturbation, which can be instance-dependent, or the noise perturbation between different subgroups can be correlated, rather than independent. Under such scenarios, we may need potential adjustment to our framework, as fairness measures under clean and noisy data can have more complicated relationships. We'll add the discussion in final paper.

[1] Rezaei, Ashkan, et al. "Fairness for robust log loss classification." Proceedings of the AAAI Conference on Artificial Intelligence. Vol. 34. No. 04. 2020.

[2] An, Bang, et al. "Transferring fairness under distribution shifts via fair consistency regularization." Advances in Neural Information Processing Systems 35 (2022): 32582-32597.

[3] Makur, Anuran, and Lizhong Zheng. "Bounds between contraction coefficients." 2015 53rd Annual Allerton Conference on Communication, Control, and Computing (Allerton). IEEE, 2015.

[4] Redmond, Michael. Communities and Crime. UCI Machine Learning Repository. (2009).

---

### Decision · Program_Chairs · 2023-09-21

**Decision:**

Reject

**Comment:**

Following the author discussion, the reviewer sentiment was generally leaning towards acceptance (with the qualifier that one positive reviewer had extremely low confidence). One dissenting reviewer raised a number of detailed concerns, centering around lack of clarity on the theory, limitation to binary labels, and confusion regarding covariate versus distribution shift. These were not fully resolved during the discussion period.

Following this, the AC read the paper as well. Based on this, we lean more towards Reviewer 5tmy's view: while the paper certainly addresses an important problem, we believe the paper could benefit from further work and review. In particular, the presentation could be significantly improved, both at the macro-level (unclear discussions of key points) and micro-level (typos, formatting issues).

Regarding the description of the main proposal --- i.e., a distribution shift approach to coping to noise in fairness problems --- there are many points that are unclear in the submitted version. Some points were made clearer in the author response. On balance, the amount of things that need to be clarified and redone exceed those that fall under a minor revision.

- L140, the statement "different noise rates can become ignorable under hyperparameter-tuning due to the proportionality" is not clear. As this justifies why prior work is not sufficient, more clarity is important.

- L142, the definition of $P$ and $Q$, which are distributions, seems to rely on $S_{ya}$, which is sample-dependent. This is a bit confusing.

- L142, L174, it is not immediately clear what is mean by the distribution of predicted soft labels. It would significantly help to spell things out more precisely. For example, presumably $P_{ya} = Pr(x | y, a)$. Does $Q_{ya} = Pr(f(x) | y, a)$?

- L174, it is not clear why minimizing the divergence between $P$ and $\hat{P}$ will also minimize the divergence between $Q$ and $\hat{Q}$. What does the latter divergence look like? (The author response included a discussion on this point.)

- L179, it is not clear why ensuring $\hat{P}\_{ya} = \hat{P}\_{ya'}$ is a valid goal. Per L142, $\hat{P}$ refers to the data distribution under noise. This is not something under the learner's control. The soft label distribution $\hat{Q}$ is more plausibly under the learner's control, but this is not mentioned here.

- the use of normalizing flows to find good representations is presented rather tersely. It is unlikely that those without deep background in this topic can follow the discussion in Section 3.3. It is not clear whether there are other approaches that are viable, and why this particular approach is to be favored. (The author response included a discussion on this point, which somewhat helps. This really needs a much more thorough discussion if it is the basis of the paper, though.)

- Equation 5, it is not clear what values $a, a'$ take here. Please make clear what arguments $L_y$ takes, as these appear to be used in Equation 6.

- Equation 6, the motivation for this form of the objective is, at a minimum, unclear.

- Equation 6, $L_{\rm cls}$ does not appear to be defined. (The author response included a discussion on this point.)

- It is not clear how Theorem 1 aligns with the proposal in Sec 3.3. Does the proposal specifically target the equalized odds measure?

- It is not clear how Lemma 2 and 3 aligns  with the proposal in Sec 3.3. Does the proposal automatically handle the case of label noise?

Further to the above, Reviewer 5tmy's comments about distribution versus covariate shift are valid, and it is suggested that a future version of the paper include a careful discussion of this point.

Other suggestions:

- Please work on reducing run-on sentences (e.g., L128, L172, L183), and fixing typographic issues (e.g., L67, incorrect placement of period in "What’s more. we extend our method", L222, "addictive constant").

- L122, it is common to use \hat{.} to denote empirical estimates. The use to denote noise may be reconsidered.

- L129, "function of classifier" is not clear.

- The use of subscript $f$ in $l_f$ presumably denotes "fairness". However there is a potential confusion with the use of $f$ as the underlying classifier.

- The equations in Sec 3.1 are hard to follow in inline format.

- L141, the paper appears to interchangeably use "attribute" and "group"; a consistent choice might be better.

- Lemma 1, please consider aligning the equations, and include the definitions of DTPR, etc., within the lemma block itself.

- Lemma 1, please include the assumptions on $\eta$ within the lemma block itself.

- Equation 3, please include the base measure for the integral.

- L187, if the method eventually uses symmetric KL, it could be clearer to start with that rather than regular KL.

- Figure 1, 2, 3: please increase the font size and line thickness.